materials science/inorganic chemistry/ environmental chemistry

TiO$_2$ thin film, spray pyrolysis, photocatalysis, air purification, superhydrophilicity, wettability

**Authors for correspondence:**
Ibrahim Dundar
e-mail: ibrahim.dundar@ttu.ee
Ilona Oja Acik
e-mail: ilona.oja@ttu.ee

One contribution to ISIEM2018 special collection.

This article has been edited by the Royal Society of Chemistry, including the commissioning, peer review process and editorial aspects up to the point of acceptance.

# TiO$_2$ thin films by ultrasonic spray pyrolysis as photocatalytic material for air purification

Ibrahim Dundar[1], Marina Krichevskaya[2], Atanas Katerski[1] and Ilona Oja Acik[1]

[1]Department of Materials and Environmental Technology, Laboratory of Thin Film Chemical Technologies, and [2]Department of Materials and Environmental Technology, Laboratory of Environmental Technology, Tallinn University of Technology, Ehitajate tee 5, 19086 Tallinn, Estonia

ID, 0000-0002-4519-7009

In this study, we showed that the TiO$_2$ thin films deposited onto window glass are practicable for air purification and self-cleaning applications. TiO$_2$ films were deposited onto window glass by ultrasonic spray pyrolysis method. Different deposition temperatures were used in the range of 250–450°C. The structural, morphological, optical properties and surface chemical composition were investigated to understand probable factors affecting photocatalytic performance and wettability of the TiO$_2$ thin films. The TiO$_2$ thin films were smooth, compacted and adhered adequately on the substrate with a thickness in the range of 100–240 nm. X-ray diffraction patterns revealed that all the TiO$_2$ thin films consisted of anatase phase structure with the mean crystallite size in the range of 13–35 nm. The optical measurements showed that the deposited films were highly transparent (approx. 85%). The wettability test results showed that the TiO$_2$ thin films sprayed at 350°C and 450°C and annealed at 500°C for 1 h were superhydrophilic. The photocatalytic activity of the films was tested for the degradation of methyl tert-butyl ether (MTBE) in multi-section plug-flow reactor. The TiO$_2$ film deposited at 350°C exhibited the highest amount of conversion of MTBE, approximately 80%.

## 1. Introduction

Volatile organic compounds (VOCs) are widespread air contaminants in both outdoors and indoors. VOCs cause serious diseases including damage to kidney, liver and central nervous system; eye, nose and throat irritation; loss of coordination and nausea. Some compounds cause cancer in animals; some are considered to cause cancer in humans [1].

Methyl tert-butyl ether (MTBE) is a VOC widely used as an octane number booster in fuels for gasoline engines and thus

contributing to the outdoor and indoor air pollution. Several studies have been done about photocatalytic decomposition of gas-phase MTBE on immobilized commercial $TiO_2$ (P25) ($TiO_2$, $TiO_2$/Pt-, S- and N-doped $TiO_2$) particles coated onto the support materials or walls of photocatalytic reactors, which are mostly annular tubular batch or continuous reactors [2–6]. In these studies, the effect of various reaction parameters such as temperature, initial MTBE concentration, oxygen concentration, residence time, relative humidity (RH) and the photocatalytic degradation of MTBE under visible light region were investigated.

$TiO_2$ exhibits a remarkable promise in the photocatalytic treatment of VOCs due to high photocatalytic activity under UV light and photostability. In addition, $TiO_2$ is chemically inert, corrosion resistant and inexpensive [7]. Generally, the immobilized form of $TiO_2$ photocatalyst has been prepared with either commercial $TiO_2$ nanopowders or thin films composed by nanosized $TiO_2$ crystals. Coatings prepared from nanopowders are less mechanically stable, i.e. have weaker adhesion to the substrate than thin films [8,9]. Moreover, the latest toxicity studies on $TiO_2$ powders indicated that $TiO_2$ nanoparticles, smaller than 20–30 nm, may cause a severe health risk [10,11]. Thus, the immobilization of $TiO_2$ with good adhesion, for example, in the form of thin films became a necessity to avoid the release of nanoparticles into the atmosphere. Nanopowder coating has low transmittance in the visible spectral range [12], which limits the field of applications as indoor or outdoor photocatalyst. $TiO_2$ thin films, however, are less studied in the gas-phase photo-oxidation due to the lower photocatalytic efficiency compared with the coatings made of commercial $TiO_2$ nanopowders [8].

To extend $TiO_2$ thin film applicability, a photocatalytic thin film must meet the following requirements: high photocatalytic activity, superhydrophilicity, high transparency, mechanical features regarding adhesion to substrate and stability against abrasion. A desirable coating technique that provides durable and stable coating, effective contact between the catalyst and the contaminant, cost-efficiency and suitability for large-scale applications [13] is needed to obtain the required material characteristics. Further, the deposition of $TiO_2$ photocatalyst onto window glass has a promising prospect for the degradation of air pollutants and self-cleaning application because window glass has significant market potential and is a cheap support material.

Different methods have been used to fabricate photoactive $TiO_2$ coatings with different characteristics for air purification such as sol–gel dip coating, spin coating [9,14], chemical vapour deposition [15], sputtering [16] and electrochemically assisted deposition [17]. Sol–gel dip coating is considered as the most common deposition method of photocatalytic $TiO_2$ coatings for the degradation of gaseous organic pollutants. However, despite the factors making it favourable such as simplicity and low-cost, it is not easy to fabricate the photocatalytic layer with desired mechanical properties [7].

Addamo et al. [9] have used sol–gel dip coating technique to deposit $TiO_2$ thin films of different thickness (100–300 nm; transparency 70%) on glass substrates and test their photocatalytic activity for degradation of gaseous 2-propanol. It was reported that the highest degradation rate was obtained on the film with the thickness value of 250 nm [9]. Ardizzone et al. [17] deposited single- and double-layer $TiO_2$ thin films on glass substrates by electrochemically assisted method. The average thickness of the films and transparency was 450 nm and 75%, respectively. $TiO_2$ thin films with double layer showed 100% ethanol (275 ppm) degradation in 120 min under UV irradiation [17].

Ultrasonic spray pyrolysis is a simple, fast, inexpensive and freely applicable method of deposition for large area coatings. Despite the easy scale-up in industry and the possibility to promptly cover large areas, to the best of our knowledge, there is a very limited number of studies about $TiO_2$ thin films deposited by ultrasonic spray pyrolysis in the literature. Da et al. [18] prepared $TiO_2$ and N-doped $TiO_2$ thin films, and Rasoulnezhad et al. [19] deposited $TiO_2$ and Fe-doped $TiO_2$ thin films on glass substrates by ultrasonic spray pyrolysis. The photocatalytic activity of coatings was studied by the degradation of methylene blue in aqueous phase under UV or visible light.

In a recent review paper on photocatalytic materials for air treatment, it has been noted that there are still a few investigations and explanations on the correlation between material properties and the photocatalytic activity towards specific VOC [20].

The present paper is a comprehensive study of unmodified $TiO_2$ thin film synthesized by ultrasonic spray pyrolysis and applied for the abatement of air pollutants. No publications on the decomposition of VOC MTBE on transparent $TiO_2$ thin films fabricated at different temperatures reporting their photocatalytic activity regarding the materials characteristics were found available, thus this study would supply more insights into this topic.

Therefore, the aim of the study was to deposit transparent TiO$_2$ thin films by ultrasonic spray pyrolysis and to investigate their structural, morphological, optical properties, surface chemical composition, wettability and photocatalytic activity toward MTBE conversion as a function of the deposition temperature. Multi-section plug-flow reactor was used for the study of photocatalytic activity of TiO$_2$ films.

# 2. Material and methods

## 2.1. Synthesis and materials characterization

TiO$_2$ thin films were deposited onto commercial window glass with a thickness of 2 mm by using the ultrasonic spray pyrolysis technique. The spray solution was composed of titanium (IV) isopropoxide (0.2 mol l$^{-1}$) and acetylacetone in a molar ratio of 1 : 4 in ethanol. The spraying rate was set up to 2.5 ml min$^{-1}$ and compressed air was applied as the carrier gas with a flow rate of 8 l min$^{-1}$. The distance of the ultrasonic nozzle to the hot plate, where the substrates were placed, was fixed at 7 cm. The number of spray cycles was set to six. The hot plate temperature was adjusted to 250°C, 350°C and 450°C, named as deposition temperature throughout the article. All as-deposited samples were annealed at 500°C for 1 h in air in a furnace Nabertherm L5/11/06D, and are named as as-prepared samples throughout the article.

X-ray diffraction (XRD) and Raman spectroscopy methods were used to investigate the structure of the samples. XRD patterns were recorded on a Rigaku Ultima IV diffractometer with Cu K$\alpha$ radiation ($\lambda = 1.5406$ Å, 40 kV at 40 mA). The measurements were carried out in 2 theta configurations with the scan range of 20–60°, with a scanning speed of 2° min$^{-1}$ and with a step of 0.02°. The mean crystallite size was calculated by the Scherrer method from the FWHM (full width at half maximum) of the (101) reflection of TiO$_2$ anatase phase. Raman spectra were obtained on a micro-Raman spectrometer HORIBA Jobin Yvon Model HR800 in the spectral range of 100–800 cm$^{-1}$. A 532 nm laser line was used for excitation, which gives 5 mW of power at 10 μm laser spot size during the measurement.

The surface morphology of the samples was investigated by scanning electron microscopy (SEM) and atomic force microscopy (AFM) measurements. The surface morphology and the film thickness of the films were obtained with the help of Zeiss HR FESEM Ultra 55 scanning electron microscopy with an acceleration voltage of 4.0 kV. The surface morphology of the films was studied using NT-MDT Solver 47 PRO atomic force microscopy system, measurement was carried out in the non-contact mode with a resolution in the range of 3 nm, and the investigated area was $1000 \times 1000$ nm per scan. The surface roughness analysis was carried out through the three-dimensional AFM scan, which was in accordance with the ISO 4287/1 standard. Water contact angle (CA) measurements were applied to investigate the wettability of the films. DSA 25 (KRÜSS Instrument) was used at room temperature, applying a sessile drop fitting method. Four spots were averaged for per substrate. UV-A irradiation in the surface wettability test was performed using Actinic BL 15 W fluorescent lamp (Philips), with max emission at 365 nm. X-ray photoelectron spectroscopy (XPS) study was performed with use of a Kratos Analytical AXIS ULTRA DLD spectrometer in conjunction with a 165 mm hemispherical electron energy analyser and delay-line detector. The analysis was carried out with monochromatic Al K$\alpha$ X-rays (1486.6 eV) operating at 15 kV and 225 W. All XPS spectra were recorded using an aperture slot of $300 \times 700$ μm and pass energy of 20 eV. Binding energy (BE) values for TiO$_2$ were calculated based on the C1s peak at 285.0 eV. The total transmittance spectra of the TiO$_2$ films on glass substrate and glass substrate as reference were measured in the 250–800 nm range using a Jasco V-670 UV-VIS-NIR spectrophotometer equipped with a 40 nm integrating sphere.

## 2.2. Evaluation of photocatalytic activity

The photocatalytic activity of thin films was studied following the photocatalytic degradation of model air pollutant MTBE (C$_5$H$_{12}$O) in gas phase. The inlet concentration of the gaseous pollutant was 10 ppm. The study of the photocatalytic activity of thin films was carried out in a multi-section plug-flow photocatalytic reactor. The multi-section reactor consists of five sections, where the section volume is 130 ml and the surface area of photocatalytic coating in one section of the reactor is 120 cm$^2$ with the overall surface area of 600 cm$^2$ in the whole reactor. Fourier transform infrared analyser (FT-IR, Interspec 200-X) with the Specac Tornado 8-m 1.33 l gas cell and in-line humidifier were assembled to

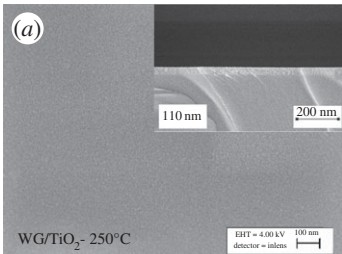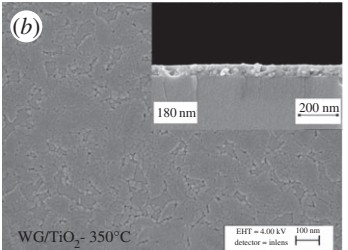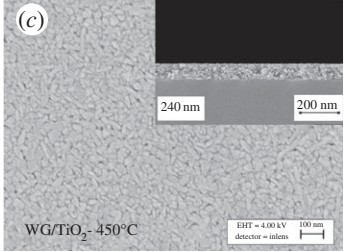

**Figure 1.** SEM surface images of TiO$_2$ thin films deposited on window glass at (a) 250°C, (b) 350°C, (c) 450°C, and annealed at 500°C for 1 h. Inset of figures (a), (b) and (c) shows the cross-sectional SEM images of the corresponding TiO$_2$ thin film.

**Table 1.** Summary of the morphological and structural properties of TiO$_2$ thin films deposited at different temperatures.

| deposition temperature (°C) | thickness (nm) | RMS (nm) | phase | mean crystallinity size (nm) | degree of crystallinity (%) |
|---|---|---|---|---|---|
| 250 | 110 | n.a. | anatase | 13 | 33 |
| 350 | 180 | 0.8 | anatase | 35 | 85 |
| 450 | 240 | 1.2 | anatase | 32 | 98 |

the reactor. A 15 W fluorescent lamp (Actinic BL, Philips) with UV-A emission intensity of 3.3 mW cm$^{-2}$ (integrated into the range of 180–400 nm, with max emission at 365 nm, UV-B/UV-A ratio less than 0.2%) was placed over each section of the reactor. The RH in the gas stream was 6% determined at 20°C and temperature in the reactor was 30°C, maintained by the heat of the lamp and controlled with the temperature controller (Omega CN9000A).

The VOC feed tank was charged with polluted air by tank evacuation and injection of a respective amount of MTBE through the injection port. After 20 min of evaporation, the tank was filled with compressed air to pressure three bars and left for the balancing of concentration fluctuations for 90 min. The gas flow controller provided a gas flow rate of 0.5 l min$^{-1}$ thus keeping the residence time of 15.6 s in the reactor section.

The MTBE peaks were measured at the IR bands from 1063 to 1124 cm$^{-1}$. The gas-phase intermediate product of photocatalytic oxidation of MTBE, tert-butyl formate (TBF), was also monitored quantitatively (at the IR bands from 1138 to 1190 cm$^{-1}$), by FT-IR outlet gas analysis using FDM VPFTIR HiRes quantitative spectra library. No other gas-phase products were observed, except carbon dioxide and water.

The reference experiments to examine the MTBE adsorption and photochemical degradation were carried out. No adsorption of MTBE on TiO$_2$ thin film catalyst in dark conditions was obtained. No photochemical decomposition of MTBE was observed under UV light in the absence of the catalyst. In both reference experiments, the initial concentration of pollutant remained unchanged (no difference between the reactor inlet and outlet) during 30 min of polluted air passing through the five sections of the reactor.

# 3. Results and discussion

## 3.1. Surface morphology

The SEM images of the TiO$_2$ films deposited onto window glass substrate at temperatures from 250°C to 450°C and annealed at 500°C for 1 h are presented in figure 1. The TiO$_2$ films are smooth, dense and free from cracks. According to the results, the surface morphology of the films deposited onto window glass changes with the increase of deposition temperature from 250°C to 450°C (figure 1a–c). The TiO$_2$ film deposited at 250°C has a plane surface structure and shows grains with a size of approximately 20 nm. The films deposited at 450°C have larger well-distinguished grains with a size of approximately 50 nm. TiO$_2$ film sprayed at 350°C has mixed surface morphology consisting of the morphology of the films deposited at 250°C and 450°C.

The thickness of all as-prepared thin films was determined from their SEM cross-sectional images. We observed that the TiO$_2$ thin films deposited on window glass have a thickness value of 110, 180 and 240 nm when grown at temperatures of 250, 350 and 450°C, respectively (table 1). The increase in the

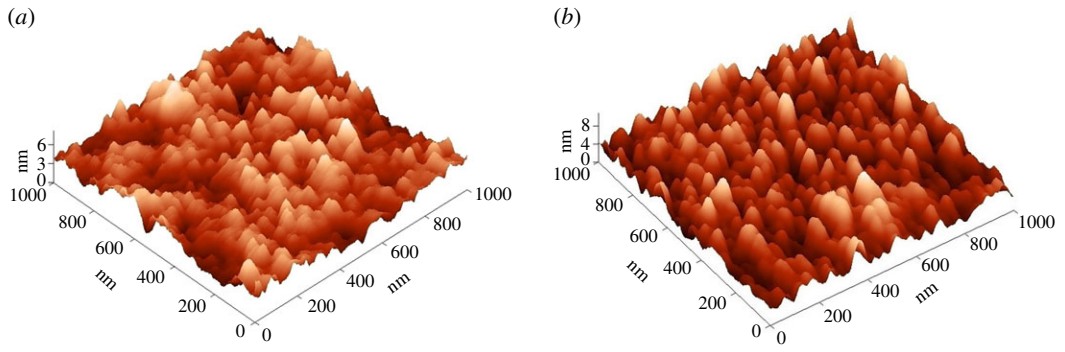

**Figure 2.** AFM images of TiO$_2$ thin films deposited on window glass at (a) 350°C and (b) 450°C and annealed at 500°C for 1 h.

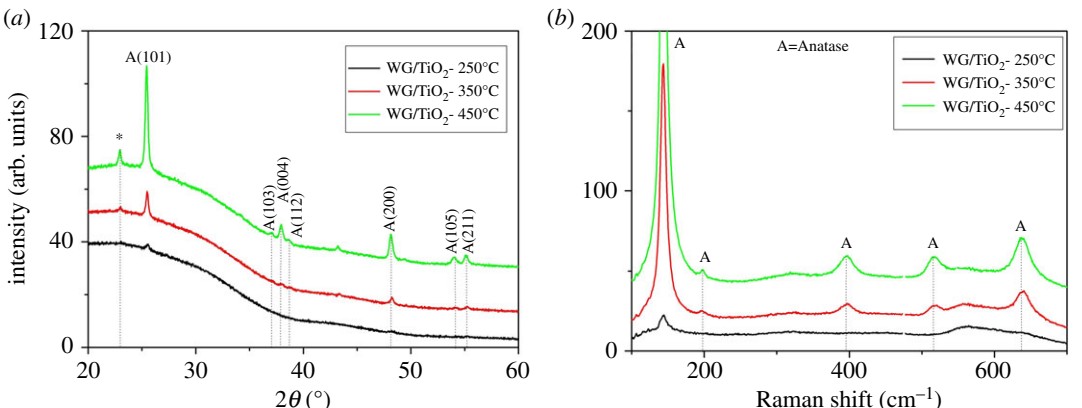

**Figure 3.** (a) XRD patterns and (b) Raman spectra of TiO$_2$ films deposited at 250°C, 350°C and 450°C onto window glass. All films were annealed at 500°C for 1 h in air.

film thickness with deposition temperature has been found also in other studies, e.g. in the case of ZrO$_2$ and TiO$_2$ deposition by spray pyrolysis [21,22].

Figure 2 shows the three-dimensional AFM deflection images of TiO$_2$ films deposited at 350°C and 450°C onto window glass and annealed at 500°C for 1 h (figure 2a,b). The surface morphology of TiO$_2$ film deposited at 350°C showed plane surface (figure 2a). The surface transforms to individually distinctive grains when increasing the deposition temperature to 450°C (figure 2b).

The root mean square (RMS) roughness was calculated from the AFM height profile of the scanned area of 1 × 1 μm. The TiO$_2$ thin film deposited at 350°C and 450°C showed RMS roughness values of 0.8 and 1.2 nm, respectively (table 1). The slightly higher RMS roughness of the TiO$_2$ film deposited at 450°C could be due to the different topography of the film and the cavities formed between the well-distinguished grains, also confirmed by SEM photos (figure 1c).

## 3.2. Structural properties

The XRD patterns of the TiO$_2$ films were analysed to obtain information on phase composition, the mean crystallite size and the degree of crystallinity. Figure 3a shows the XRD patterns of TiO$_2$ films deposited at 250°C, 350°C and 450°C onto window glass substrates and annealed at 500°C for 1 h in air. As shown in figure 3a, the XRD patterns of TiO$_2$ films exhibited diffraction peaks at 2 theta of 25.5°, 37.8°, 48.2°, 53.9° and 55°, which belong to the TiO$_2$ anatase structure [23]. Additional diffraction peak appeared at 2 theta of 22.9° belongs to SiO$_2$ from the substrate. No diffraction peaks corresponding to rutile or brookite phase of TiO$_2$ were observed. The mean crystallite size of TiO$_2$ was calculated from the (101) peak of anatase phase by the Scherrer formula. The mean crystallite size of films on window glass increased in the range of 13–35 nm with the increase of deposition temperature from 250°C to 450°C (table 1).

The degree of crystallinity of the TiO$_2$ thin films was determined by calculating crystalline/ amorphous ratios based on scattered intensity (I) of anatase at 2 theta of 25.5° and 48.2° and intensity

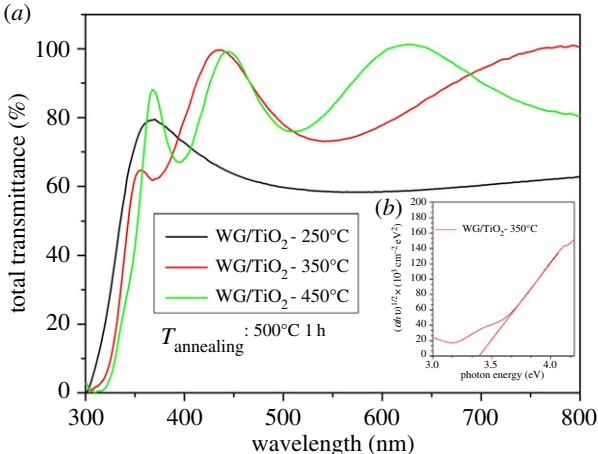

**Figure 4.** (*a*) Total transmittance spectra for $TiO_2$ thin films deposited at 250°C, 350°C and 450°C onto window glass and annealed at 500°C for 1 h. (*b*) The inset shows the band gap value of as-prepared $TiO_2$ sample deposited at 350°C.

from amorphous phase. The peak intensity of the amorphous phase was detected after applying profile fitting used for the separation of amorphous phase from crystalline phase (equation (3.1)) [24]

$$Crystallinity, \% = \frac{I\ crystalline\ phase}{I\ crystalline\ phase\ +\ I\ amorphous\ phase}. \qquad (3.1)$$

The degree of crystallinity of the $TiO_2$ film grew from 33% to 98% with the deposition temperature increased from 250°C to 450°C (table 1). This confirms that the crystallization evolution of $TiO_2$ films is determined by the deposition temperature despite all samples being annealed at the same temperature. Similar behaviour, where the crystallinity of the film was controlled by both the deposition and annealing temperature, has been observed also for $TiO_2$ thin films deposited by pulsed pneumatic spray pyrolysis method [22].

Figure 3*b* shows the Raman spectra of the $TiO_2$ films deposited at different temperatures at 250°C, 350°C and 450°C onto window glass and annealed at 500°C for 1 h. The Raman spectra showed bands at 143 (*Eg*), 197 (*Eg*), 396 (*B1g*) and 637 (*Eg*) cm$^{-1}$, which are characteristic of $TiO_2$ anatase phase with no peaks belonging to the rutile or brookite $TiO_2$ phase, thereby confirming the XRD results.

## 3.3. Optical properties

The optical transmittance spectra of $TiO_2$ thin films deposited at different temperatures and annealed at 500°C were measured in the wavelength range between 250 and 800 nm. The total transmittance of the $TiO_2$ film deposited at 350°C and 450°C is approximately 85% in the spectral region of 400–800 nm (figure 4*a*). The $TiO_2$ film deposited at 250°C showed lower optical transmittance compared with the films deposited at higher temperatures, which could be attributed to the differences in the nature of microstructure, thicknesses and surface morphology of the $TiO_2$ films.

Equation (3.2) was applied to determine absorption coefficient ($\alpha$) to calculate the optical band gap value, where $T$ and $d$ represent the total transmittance of the film and the thickness of the film, respectively.

$$\alpha = \frac{\ln(1/T)}{d}. \qquad (3.2)$$

The band gap $Eg$ was calculated using $T_{auc}$ equation; $k$ is constant, $Eg$ is band gap ($n = 1/2$ for indirect or $n = 2$ for direct transitions) and $h\upsilon$ is the photon energy

$$\alpha = \frac{k(h\upsilon - Eg)^n}{h\upsilon}. \qquad (3.3)$$

Figure 4*b* presents the band gap values determined by extrapolating the linear region of the plot of $(ah\upsilon)^{1/2}$ against the photon energy to detect the indirect band gap values. $TiO_2$ thin films deposited at 250°C, 350°C and 450°C showed $Eg$ values of 3.53, 3.38 and 3.41 eV, respectively. The optical band

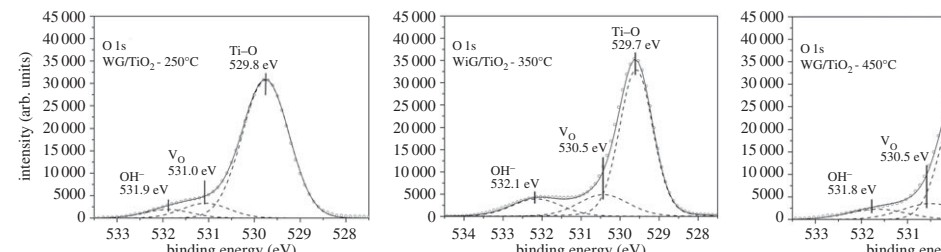

**Figure 5.** XPS spectra of as-prepared TiO₂ thin films deposited onto window glass substrates in the BE region of O1s.

gap value of bulk TiO₂ is 3.20 eV. The higher band gap value than that of the bulk TiO₂ has been observed in the different studies of the TiO₂ thin films deposited by spray pyrolysis methods [25,26].

## 3.4. Chemical composition and wettability

XPS studies were performed to investigate the chemical composition and bonding structure of the as-prepared TiO₂ thin films. Figure 5 shows the oxygen (O1s) core level spectra of the surface of TiO₂ films deposited at different temperatures and annealed at 500°C.

Lorentzian–Gaussian (function pseudo-Voigt) fitting analysis were used to deconvolute the asymmetric O1s core level peaks. Shirley type fitting was used as the background subtraction. In the O1s core level spectra of the as-prepared TiO₂ films (figure 5), the peak observed at BE value of $529.7 \pm 0.3$ eV is attributed to Me−O bond, thereby, it is ascribed to the presence of Ti−O bond. The peak observed at BE value of $530.5 \pm 0.5$ eV could be attributed to the presence of oxygen vacancies (Vo) [27,28]. The shoulder peak located at BE value of $532.1 \pm 0.3$ eV reveals that the surface of the TiO₂ film involves surface hydroxyl groups (OH⁻) [28,29].

Atomic concentrations of the components such as Ti−O, Vo and OH⁻ found in the O1s core level spectrum were calculated from the integrated areas of O1s core level spectra by using Snowfields' cross-sections. Atomic ratios of the components [OH]/[Ti−O] and [Vo]/[Ti−O] are presented in table 2.

According to the [OH]/[Ti−O] ratios (table 2), the amount of OH⁻ groups on the as-prepared TiO₂ films on window glass increased from 0.06 up to 0.18 at% with the increase in the deposition temperature from 250°C to 350°C. However, a slight drop in the [OH]/[Ti−O] ratio was observed for the as-prepared TiO₂ film deposited at 450°C. A decrease in the amount of OH⁻ groups was observed in TiO₂ thin films deposited by sol−gel methods: Simonsen *et al.* [30] reported a decrease in OH⁻ groups on the microwave assisted sol−gel TiO₂ thin films with the increase in calcination temperature. Chen *et al.* [28] studied TiO₂ thin film grown by sol−gel spin coating and reported an increase in the number of hydroxyl groups with an increase of film thickness; but with a further growth of thickness a decrease in the amount of OH⁻ groups on the film surface was observed. Ennaceri *et al.* [31] reported gradual increase in the amount of surface OH⁻ groups with the increase in the thickness of the TiO₂ thin films deposited by ultrasonic spray pyrolysis. The results of the present study disagree with the observation study of Ennaceri *et al.* [31], because the TiO₂ thin film deposited at 450°C is thicker than the sample deposited at 350°C, showing the high influence of deposition temperature on the surface chemical features. Furthermore, following the [Vo]/[Ti−O] ratios, the amount of oxygen vacancies on the surface of TiO₂ films increased with the increase in the deposition temperature from 250°C to 350°C (table 2). It was found that the highest amount of oxygen vacancies on the TiO₂ film surface belongs to the film fabricated at 350°C (table 2). The lower number of oxygen vacancies defects on TiO₂ thin film deposited at 450°C could be attributed to the higher deposition temperature: the intensive heat treatment could repair the oxygen vacancy defects. Liu *et al.* [32] studied TiO₂ thin films deposited by ultrasonic spray pyrolysis and annealed at different temperatures; a decrease in oxygen surface defects on TiO₂ thin film with an increase in annealing temperature was reported.

Thus, XPS results showed that the film deposited at 350°C has the highest amount of OH⁻ groups and oxygen vacancy defects and less carbon content on the surface of the film (table 2). The OH⁻ on TiO₂ surface are considered very effective for photocatalytic degradation of organic pollutants as they are generally the precursors of hydroxyl radicals. In addition, the surface oxygen vacancy defects are crucial because water can dissociate on oxygen vacancies with the formation of two bridging OH⁻ groups creating more OH⁻ groups on the film surface, which is beneficial for adsorbing VOCs by forming hydrogen bonds with functional groups [28,33]. The carbon impurities were found giving an

**Table 2.** Results of the XPS and the water CA studies of as-prepared TiO$_2$ thin films deposited onto window glass at different temperatures.

| XPS study of as-prepared samples | | | | | CA value | | | |
|---|---|---|---|---|---|---|---|---|
| deposition temperature (°C) | Na (at%) | C (at%) | [Vo]/ [Ti−O] (at%) | [OH]/ [Ti−O] (at%) | as-prepared | | UV treatment (30 min) | aged (three months) |
| 250 | 13.0 | 18.3 | 0.10 | 0.06 | ~33° | | ~ 7° | ~ 17° |
| 350 | 10.0 | 6.3 | 0.23 | 0.18 | ~7° | | ~0° | ~ 10° |
| 450 | 11.0 | 9.8 | 0.13 | 0.13 | ~6° | | ~0° | ~18° |

inhibiting effect to the active sites on the surface of TiO$_2$ thin films, and thus decreasing the photocatalytic activity [14].

Na$^+$ content was analysed by XPS to estimate Na$^+$ diffusion from the substrate to the film surface (table 2). According to XPS data (table 2), the amount of Na$^+$ on the surface of TiO$_2$ films is 10–13 at%. Na$^+$ content decreases with the increase in deposition temperature, which can be attributed to the higher film thickness, inhibiting diffusion of Na$^+$ ions to the surface. The detrimental effect of Na$^+$ diffusion from window glass to the film surface during the heat treatment was investigated in several studies. In this respect, different negative influences of Na$^+$ diffusion on TiO$_2$ film properties have been proposed: disorder of crystallinity of TiO$_2$ [34,35], prevention of the formation of the anatase phase [36], the production of recombination centres for photogenerated electron−hole pairs [34,35].

Surface wettability of the TiO$_2$ films deposited at various temperatures was tested by measuring the water CA. Table 2 shows the average water CA values of the as-prepared, UV-treated and aged TiO$_2$ thin films. All as-prepared samples were stored in the plastic boxes for three months.

The surface wettability of as-prepared TiO$_2$ thin films depends on the deposition temperature. The film deposited at 250°C shows water CA value of around 33°, whereas the water CA value of the samples deposited at 350°C and 450°C is below 10°, confirming the superhydrophilic nature of the samples [37]. The superhydrophilic nature of the films deposited at 350°C and 450°C could be attributed to the higher content of OH$^-$ and Vo on the film surface compared with the film deposited at 250°C (table 2). It is commonly reported in the literature that as the OH$^-$ content on the film surface increases, van der Waals and hydrogen bonding between the OH$^-$ groups and water occurs, which is expected to augment the hydrophilicity, i.e. to enhance of the wetting [28,37]. Additionally, the presence of the surface oxygen defects enhance the wetting properties, leading to the trapping of OH$^-$ groups [28]. Simonsen *et al.* [30] studied the effect of OH$^-$ groups on superhydrophilic properties of TiO$_2$ thin films with different characteristics: a linear correlation between the OH$^-$ amount on the surface of the film and the water CA was observed. On the other hand, Chen *et al.* reported that the TiO$_2$ thin film which has the lowest level of OH$^-$ and Vo, also demonstrated the superhydrophilic property under UV light [28].

After UV treatment for 30 min, the water CA of all as-prepared samples drops below 10°, indicating superhydrophilic behaviour. Moreover, the UV-treated samples deposited at 350°C and 450°C showed the water CA value of 0° irrespective of the deposition temperature. Most often, the enhancement of wetting properties of TiO$_2$ surface under UV irradiation was attributed to the creation of photo-induced oxygen vacancies [28,37], reconstruction of the surface hydroxyl groups [37,38] and photo-induced removal of carbon residues on TiO$_2$ surface [37,39].

The UV-treated samples were aged by storing them in the plastic box for three months. The increase in water CA irrespective of the deposition temperature was observed (table 2), whereas the adsorption of carbon residues on the surface of TiO$_2$ films in air could be the reason of this tendency [40]. TiO$_2$ films deposited at 350°C exhibited less changes in water CA over time, compared to the other TiO$_2$ films. The

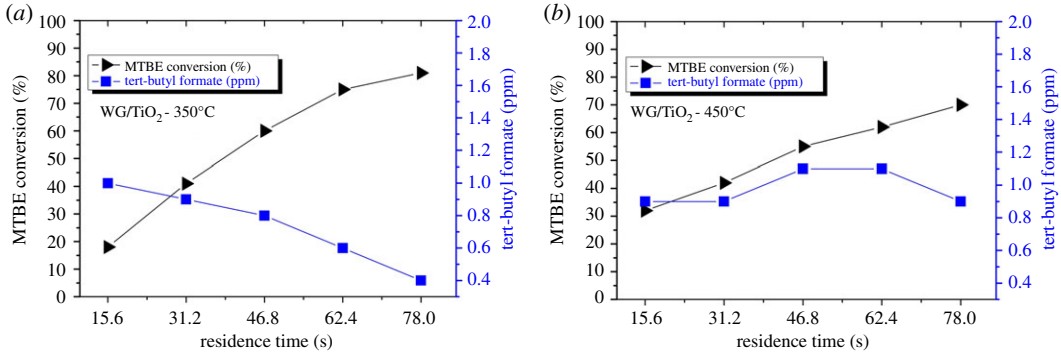

**Figure 6.** Photocatalytic conversion of MTBE on $TiO_2$ thin films deposited onto window glass at (*a*) 350°C and (*b*) 450°C. All samples were annealed at 500°C; MTBE inlet concentration 10 ppm; RH 6%; reactor temperature 30°C.

$TiO_2$ films deposited on window glass at 350°C maintained their superhydrophilicity, i.e. a water CA of 10° even after a storage in a plastic box for three months.

## 3.5. Photocatalytic activity

Photocatalytic activity of $TiO_2$ thin films was studied following the photocatalytic gas-phase degradation of MTBE in the multi-section continuous plug-flow reactor at operating conditions kept constant for all the experimental runs: residence time 15.6 s per section, RH 6% and reactor temperature 30°C.

In different studies, acetone, TBF ($C_5H_{10}O_2$), isobutene and tert-butyl alcohol have been detected as the intermediate products of gas-phase photocatalytic degradation of MTBE apart from final oxidation products $CO_2$ and $H_2O$ [4,41]. In this study, the only gas-phase intermediate product observed during the degradation of MTBE was TBF. Other intermediates presumably formed during further degradation of MTBE or TBF were not detected in gas phase; either they were adsorbed and degraded on the catalytic surface or desorbed in amounts lower than the detection limit of the analytical apparatus (less than 500 ppb).

The photocatalytic oxidation of MTBE characterized by its conversion defined as $(C_{in}-C_{out})/C_{in}$ (%) and the formation of TBF, are illustrated in figure 6 as a function of residence time which was increased by subsequent inclusion of reactor sections. Except for $TiO_2$ film deposited at 250°C, the degradation of MTBE was examined at five residence times, 15.6, 31.2, 46.8, 62.4 and 78.0 s, using one, two, three, four or all five sections of the reactor, respectively. The photocatalytic activity of $TiO_2$ thin films was first studied using one section of the reactor, i.e. 120 cm$^2$ of photocatalytic coating, where the film deposited at 250°C showed the lowest MTBE conversion of (approx. 4%). Thereafter, this was not synthesized in an amount needed for more extensive study. The lower activity of the $TiO_2$ film deposited at 250°C compared with the films deposited at higher deposition temperature could be attributed to the low crystallinity (table 1) and low $OH^-$ content on the film surface (table 2).

A coating area of 600 cm$^2$ was applied in the study of photocatalytic activity of the films deposited at 350°C and 450°C. Figure 6 shows MTBE conversions, which, as expected, are higher at longer residence time.

$TiO_2$ thin film deposited at 350°C exhibited the highest MTBE conversion of approximately 80%. It should be emphasized that the $TiO_2$ film deposited at 450°C (approx. 70%) is less active than the $TiO_2$ film grown at 350°C despite the higher crystallization and smaller crystallite size (table 1). This could be explained by the mixed surface morphology of the sample deposited at 350°C if compared to the films deposited at 250°C and 450°C (figure 1) as well as by the higher amount of hydroxyl groups and oxygen defects (see §3.4).

Different stages of typical intermediate formation profile could be observed during the degradation of MTBE on $TiO_2$ photocatalytic thin films. The amount of TBF formed on the $TiO_2$ film deposited at 350°C is consistently decreasing (at 78 s only 5% of degraded MTBE appeared as gaseous TBF). Thus the rate of by-product degradation is higher than its formation rate (figure 6*a*). With the film deposited at 450°C on window glass, the trend towards TBF degradation is observed if residence time is longer than 62 s (about 13% of degraded MTBE formed TBF at 78 s of residence time). There have been several studies revealing that many factors can affect the formation of intermediates in the photodegradation reactions such as type of the catalyst and its amount, the technique used, etc. [41,42].

In the present study, the thickness of $TiO_2$ thin films increased with an increase to the deposition temperature (table 1). It has been reported [43] that higher photocatalytic activity is attributed to a rougher surface of the film with higher thickness. However, in several studies, it has been reported that with the further increase in the thickness of films, the length of the migration path of the carriers to the surface of the catalyst increases while their generation rate remains constant; the charge carriers experience higher recombination rates, resulting in an overall decrease in the photocatalytic activity [43]. This could be another reason of lower photocatalytic activity of the $TiO_2$ thin film deposited at $450°C$ if compared to the $TiO_2$ film deposited at $350°C$.

Usually, it is quite difficult to compare the photocatalytic performance of thin films with that of coatings obtained from nanoparticles. The specific quantity of $TiO_2$ expressed as mg of titania per $cm^2$ of surface area of the coating could serve for a proper comparison of different studies regarding the photocatalytic oxidation of MTBE. In the present study, the specific quantity of $TiO_2$ in the film deposited at $350°C$ was around $0.2\ mg\ cm^{-2}$, achieving 80% conversion of MTBE on a coated window glass with the surface area of $600\ cm^2$. Preis et al. [6] reported the specific quantity of $TiO_2$ (P25) in the coated reactor ($640\ cm^2$) as $1.4\ mg\ cm^{-2}$. The 30% conversion of MTBE (inlet concentration of 100 ppm) with acetone detected as the only gas-phase by-product was observed in this reactor (the reactor temperature was $59.8°C$). Galanos et al. [5] indicated the specific quantity of $TiO_2$ in the coated reactor as $3.5\ mg\ cm^{-2}$. This $TiO_2$ coating showed 90% of MTBE (500 ppm) conversion in up to 36.6 s of residence time. Acetone and TBF were the only intermediate by-products detected during the photocatalytic reaction. Park et al. [4] presented the calculated reaction rate values during the photocatalytic degradation of MTBE as a function of different specific quantities of $TiO_2$. It has been found that the highest reaction rate ($0.85\ \mu mol\ min^{-1}$) was obtained with the $TiO_2$ (P25) coating of $0.6\ mg\ cm^{-2}$ and that the reaction rate stayed constant even after the increase in the specific catalyst quantity.

# 4. Conclusion

Highly transparent, smooth and crack-free $TiO_2$ thin films were deposited onto commercially available window glass by ultrasonic spray pyrolysis method in the temperature range of $250–450°C$, followed by annealing at $500°C$ for 1 h in air. The effect of the deposition temperature on the morphological, structural, optical and surface chemical composition was examined to comprehend the factors affecting wettability and photocatalytic activity of the deposited $TiO_2$ films. According to SEM cross-sectional images, the thickness of the $TiO_2$ films increased from 110 to 240 nm with the increase of the deposition temperature from $250°C$ to $450°C$. According to XRD, all as-prepared $TiO_2$ films possess anatase crystalline structure, and the degree of crystallinity increased from 33% to 98% with the increase of the deposition temperature from $250°C$ to $450°C$. The mean crystallite size of the films was found to depend on the deposition temperature and remained in the range of 13–35 nm. Surface wettability test showed that as-prepared $TiO_2$ thin films sprayed at $350°C$ and $450°C$ are superhydrophilic [$CA \leq 10°$], showing CA values of $7°$ and $6°$, respectively, and CA value of $0°$ after 30 min UV treatment. The surface of the film deposited at $350°C$ remained superhydrophilic even after ageing for three months. According to the XPS study, the amount of oxygen defects and $OH^-$ groups on the $TiO_2$ film surface depends on the deposition temperature. The $TiO_2$ thin film deposited at $350°C$ exhibited the highest amount of oxygen defects and $OH^-$ groups on the film surface, and high amount of $Na^+$ (10 at%).

This study showed, that transparent and superhydrophilic $TiO_2$ thin films with the specific quantity of $0.2\ mg\ cm^{-2}$ and surface area of $600\ cm^2$ is effective to degrade approximately 80% of MTBE and its intermediate product TBF; and is thereby prospective coating for self-cleaning and air purification applications.

Data accessibility. Data available from the Dryad Digital Repository: https://datadryad.org/resource/doi:10.5061/dryad. b24p647 [44].

Authors' contributions. I.D. carried out the material synthesis and material characterizations (XRD, Raman spectroscopy, optical properties and wettability test), participated in data collection for photocatalytic activity test, participated in the design of the all figures and tables, and participated in writing of manuscript and drafted the manuscript. M.K. carried out photocatalytic activity test, interpretation of data and participated in writing of manuscript. I.O.A. carried out the design of the study, participated in data analysis and writing of the manuscript. A.K. participated in synthesis and made XPS data analysis. All authors gave final approval for publication.

Competing interests. We declare we have no competing interests.

Funding. I.D. is supported by the Estonian Ministry of Education and Research, Estonian Research Council project IUT19-4, Estonian Centre of Excellence project TK141 and EU regional Fund project 2014-2020.4.01.16-0032. I.O.A. and A.K. are supported by the Estonian Ministry of Education and Research, Estonian Research Council projects IUT19-4 and Estonian Centre of Excellence project TK141. M.K. is supported by the Estonian Ministry of Education and Research, Estonian Research Council projects IUT1-7. This work has been partially supported by ASTRA 'TUT Institutional Development Programme for 2016–2022' Graduate School of Functional Materials and Technologies (2014-2020.4.01.16-0032).

Acknowledgments. The authors acknowledge Dr V. Mikli for SEM, Dr M. Danilson for XPS measurements and Mr A. O. Titilope for AFM measurements.

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
