## [Reviewer comments · Royal Society Open Science]

Review History

RSOS-181578.R0 (Original submission)

Review form: Reviewer 1

Is the manuscript scientifically sound in its present form?

Yes

Are the interpretations and conclusions justified by the results?

Yes

Is the language acceptable?

Yes

Is it clear how to access all supporting data?

Yes

Do you have any ethical concerns with this paper?

No

Have you any concerns about statistical analyses in this paper?

No

Recommendation?

Accept as is

Comments to the Author(s)

The authors of the paper entitled: "TiO₂ thin films by ultrasonic spray pyrolysis as photocatalytic material for air purification", presented the study of deposited TiO₂ thin films by ultrasonic spray pyrolysis method. The scientific motivation has been presented in a satisfactory way. The structural, morphological, optical properties and surface chemical composition were investigated to understand probable factors affecting photocatalytic performance for the degradation of methyl tert-butyl ether in multi-section plug-flow reactor and wettability of the TiO₂ thin films. The novelty of the investigation was the study of chemical composition and wettability of the films, as is not usual in reports on chemically sprayed TiO₂ thin films. The Introduction is clearly writing, as well as the Experimental Part. Results were clearly discussed showing consistency with other reports.

The paper is very interesting to me as well as found novelty of research finding. The manuscript is also well organized and I think it's suitable for the publication on Royal Society Open Science.

Review form: Reviewer 2

Is the manuscript scientifically sound in its present form?

Yes

Are the interpretations and conclusions justified by the results?

Yes

Is the language acceptable?

Yes

Is it clear how to access all supporting data?

Yes

Do you have any ethical concerns with this paper?

No

Have you any concerns about statistical analyses in this paper?

No

Recommendation?

Major revision is needed (please make suggestions in comments)

Comments to the Author(s)

Authors, the paper requires major changes with the better explanation. Here are my suggestions, questions, and confusion regarding your submission.

I recommend accepting the paper after a major revision.

1- In the introduction of the manuscript, the authors have not given any information about the studies carried by the other researchers which are similar with their study and also they have not mentioned at all about the difference of their study compared to the other similar studies. So,

- they must tell this difference clearly.
- 2- What is the novelty and new contribution of this paper?
 - 3- Authors should absolutely specify the new results reported in this work compared to other results.
 - 4- More and proper discussion about the results is necessary.
 - 5- Explain the increase and the decrease of grain size.
 - 6- How much the thickness values of the films?
 - 7- The English grammar should be reviewed.

Decision letter (RSOS-181578.R0)

21-Nov-2018

Dear Mr DUNDAR:

Title: TiO₂ thin films by ultrasonic spray pyrolysis as photocatalytic material for air purification

Manuscript ID: RSOS-181578

The editor assigned to your manuscript has now received comments from reviewers. We would like you to revise your paper in accordance with the referee and Subject Editor suggestions which can be found below (not including confidential reports to the Editor). Please note this decision does not guarantee eventual acceptance.

Please submit your revised paper before 14-Dec-2018. Please note that the revision deadline will expire at 00.00am on this date. If we do not hear from you within this time then it will be assumed that the paper has been withdrawn. In exceptional circumstances, extensions may be possible if agreed with the Editorial Office in advance. We do not allow multiple rounds of revision so we urge you to make every effort to fully address all of the comments at this stage. If deemed necessary by the Editors, your manuscript will be sent back to one or more of the original reviewers for assessment. If the original reviewers are not available we may invite new reviewers.

Please also include the following statements alongside the other end statements. As we cannot publish your manuscript without these end statements included, if you feel that a given heading is not relevant to your paper, please nevertheless include the heading and explicitly state that it is not relevant to your work.

• Ethics statement

Please clarify whether you received ethical approval from a local ethics committee to carry out your study. If so please include details of this, including the name of the committee that gave consent in a Research Ethics section after your main text. Please also clarify whether you received informed consent for the participants to participate in the study and state this in your Research Ethics section.

OR

Please clarify whether you obtained the necessary licences and approvals from your institutional animal ethics committee before conducting your research. Please provide details of these licences and approvals in an Animal Ethics section after your main text.

OR

Please clarify whether you obtained the appropriate permissions and licences to conduct the fieldwork detailed in your study. Please provide details of these in your methods section.

RSC Associate Editor:
Comments to the Author:
(There are no comments.)

RSC Subject Editor:
Comments to the Author:
(There are no comments.)

Reviewers' Comments to Author:
Reviewer: 1

Comments to the Author(s)

The authors of the paper entitled: "TiO2 thin films by ultrasonic spray pyrolysis as photocatalytic material for air purification", presented the study of deposited TiO2 thin films by ultrasonic spray pyrolysis method. The scientific motivation has been presented in a satisfactory way. The structural, morphological, optical properties and surface chemical composition were investigated to understand probable factors affecting photocatalytic performance for the degradation of

methyl tert-butyl ether in multi-section plug-flow reactor and wettability of the TiO₂ thin films. The novelty of the investigation was the study of chemical composition and wettability of the films, as is not usual in reports on chemically sprayed TiO₂ thin films. The Introduction is clearly writing, as well as the Experimental Part. Results were clearly discussed showing consistency with other reports.

The paper is very interesting to me as well as found novelty of research finding. The manuscript is also well organized and I think it's suitable for the publication on Royal Society Open Science.

Reviewer: 2

Comments to the Author(s)

Authors, the paper requires major changes with the better explanation. Here are my suggestions, questions, and confusion regarding your submission.

I recommend accepting the paper after a major revision.

1- In the introduction of the manuscript, the authors have not given any information about the studies carried by the other researchers which are similar with their study and also they have not mentioned at all about the difference of their study compared to the other similar studies. So, they must tell this difference clearly.

2- What is the novelty and new contribution of this paper?

3- Authors should absolutely specify the new results reported in this work compared to other results.

4- More and proper discussion about the results is necessary.

5- Explain the increase and the decrease of grain size.

6- How much the thickness values of the films?

7- The English grammar should be reviewed.

Author's Response to Decision Letter for (RSOS-181578.R0)

See Appendix A.

Decision letter (RSOS-181578.R1)

21-Jan-2019

Dear Mr DUNDAR:

Title: TiO₂ thin films by ultrasonic spray pyrolysis as photocatalytic material for air purification

Manuscript ID: RSOS-181578.R1

It is a pleasure to accept your manuscript in its current form for publication in Royal Society Open Science. The chemistry content of Royal Society Open Science is published in collaboration with the Royal Society of Chemistry.

RSC Associate Editor
Comments to the Author:
(There are no comments.)

Reviewer(s)' Comments to Author:

Appendix A

Journal: Royal Society Open Science

Manuscript Title: TiO₂ thin films by ultrasonic spray pyrolysis as photocatalytic material for air purification

Authors: Ibrahim Dundar, Marina Krichevskaya, Atanas Katerski, Ilona Oja Acik

Manuscript ID: RSOS-181578

Correction Report

The authors are very grateful for the valuable feedback from the referees and the editorial team.

Changes to the manuscript are highlighted with grey colour.

Reviewers' Comments to Author:

Review report: 1

Comments to the Author(s)

The authors of the paper entitled: "TiO₂ thin films by ultrasonic spray pyrolysis as photocatalytic material for air purification", presented the study of deposited TiO₂ thin films by ultrasonic spray pyrolysis method. The scientific motivation has been presented in a satisfactory way. The structural, morphological, optical properties and surface chemical composition were investigated to understand probable factors affecting photocatalytic performance for the degradation of methyl tert-butyl ether in multi-section plug-flow reactor and wettability of the TiO₂ thin films. The novelty of the investigation was the study of chemical composition and wettability of the films, as is not usual in reports on chemically sprayed TiO₂ thin films. The Introduction is clearly writing, as well as the Experimental Part. Results were clearly discussed showing consistency with other reports.

The paper is very interesting to me as well as found novelty of research finding. The manuscript is also well organized and I think it's suitable for the publication on Royal Society Open Science.

Review report: 2

Comments to the Author(s)

Authors, the paper requires major changes with the better explanation. Here are my suggestions, questions, and confusion regarding your submission.

I recommend accepting the paper after a major revision.

Comment 1: In the introduction of the manuscript, the authors have not given any information about the studies carried by the other researchers which are similar with their study.

Answer: The authors would like to thank the referee for his/her valuable remark. The introduction of the manuscript was improved accordingly.

Added text:

It now reads in the revised version on:

Page 2, 5th paragraph:

“Different methods have been used to fabricate photoactive TiO₂ thin films for air purification such as sol-gel dip coating, spin coating, chemical vapor deposition sputtering and, electrochemically assisted deposition. Additionally, several studies have been done about photocatalytic decomposition of gas-phase MTBE on immobilized commercial TiO₂ (P25) particles coated onto photocatalytic reactors, which are mostly annular tubular batch reactor or continuous reactors. However, a limited number of chemical deposition methods have been used to fabricate unmodified and transparent TiO₂ thin films for photocatalytic oxidation of gaseous organic pollutants.”

Page 2, 6th paragraph:

“Paola et al. [4] have used sol-gel dip coating technique to deposit TiO₂ thin films of different thickness (100-300 nm; transparency 70%) on glass substrates and test their photocatalytic activity for degradation of gaseous 2-propanol. It was reported that the highest degradation rate was obtained on the film with the thickness value of 250 nm [8]. Ardizzone et al. [17] deposited single and double layer TiO₂ thin films on glass substrates by electrochemically assisted method. The average thickness of the films and transparency was 450 nm and 75%, respectively. TiO₂ thin films with double layer showed 100% ethanol (275 ppm) degradation in 120 min under UV irradiation [17].”

Page 2, 7th paragraph:

“Ultrasonic spray pyrolysis is a simple, fast, inexpensive and freely applicable method of deposition for large area coatings. Despite the easy scale up in industry and the possibility to promptly cover large areas, to the best of our knowledge, there is a very limited number of studies about TiO₂ thin films deposited by ultrasonic spray pyrolysis in the literature. Da et al. [18] prepared TiO₂ and N-doped TiO₂ thin films and Rasoulnezhad et al. [19] deposited TiO₂ and Fe-doped TiO₂ thin films on glass substrates by ultrasonic spray pyrolysis.

The photocatalytic activity of coatings was studied by the degradation of methylene blue in aqueous phase under UV or visible light.”

Comment 2: The authors would like to thank the referee for his/her valuable remark. The introduction of the manuscript was improved accordingly.

Answer:

In the present study, TiO₂ thin films were composed by sols derived from titanium tetraisopropoxide precursor. Films showed high transparency in visible spectral region. The study of the photocatalytic activity of thin films was carried out in a multi-section plug-flow photocatalytic reactor. Besides, the TiO₂ thin films were deposited by the ultrasonic spray pyrolysis method, which was used before only to test the photocatalytic activity of TiO₂ thin film for dye degradation in water treatment applications. On the other hand, we investigated wettability behavior of TiO₂ thin films, which is an important property of the window coatings for outdoor air treatment.

Added text:

It now reads in the revised version on:

Page 2, 9th paragraph:

“The present paper is a comprehensive study of unmodified TiO₂ thin film synthesized by ultrasonic spray pyrolysis and applied for the abatement of air pollutants. No publications on the decomposition of VOC MTBE on transparent TiO₂ thin films fabricated at different temperatures reporting their photocatalytic activity regarding the materials characteristics were found available, thus this study would supply more insights into this topic.”

Comment 3: What is the novelty and new contribution of this paper?

Answer:

The combination of following items is considered as novel and original:

- The first paper on transparent TiO₂ thin films (<200nm) for photocatalytic degradation of MTBE.
- The method of film deposition, which is ultrasonic spray pyrolysis.
- The unique reactor for photocatalytic activity test, which is multi-section plug-flow photocatalytic reactor.

Comment 4:

Authors should absolutely specify the new results reported in this work compared to other results.

Answer:

The new results compared to TiO₂ thin films deposited by wet-chemical deposition methods are;

- a) Ultrasonic spray pyrolysis proved, as a promising technique for the deposition of TiO₂ thin films for air purification applications.
- b) Multi-section plug-flow reactor is highly convenient to test photocatalytic activity of TiO₂ coatings by following the degradation of gaseous organic pollutants.
- c) The results of photocatalytic activity study showed the high influence of the deposition temperature on the specific surface properties of the TiO₂ thin films deposited by ultrasonic spray pyrolysis.
- d) The wettability results of TiO₂ thin films, which exhibited superhydrophilic behaviour even for the aged samples.

Comment 5: More and proper discussion about the results is necessary.

Answer: The authors have made the following amendments to the results and discussion section:

Added texts:

It now reads in the revised version on:

4.1. Surface Morphology:

Page 4, 2nd paragraph:

“The increase in the film thickness with deposition temperature has been found also in other studies, e.g. in case of ZrO₂ and TiO₂ deposition by spray pyrolysis [21, 22].”

4.2. Structural Properties

Page 5, 3rd paragraph:

“Similar behaviour, where the crystallinity of the film was controlled by both the deposition and annealing temperature, has been observed also for TiO₂ thin films deposited by pulsed pneumatic spray pyrolysis method [22].”

4.4. Chemical Composition and Wettability

Page 7, 4th paragraph:

“A decrease in the amount of OH⁻ groups was observed in TiO₂ thin films deposited by sol-gel methods: Simonsen et al. [30] reported a decrease in OH⁻ groups on the microwave assisted sol-gel TiO₂ thin films with the increase in calcination temperature. Chen et al. [28] studied TiO₂ thin film grown by sol-gel spin coating and reported an increase of in the number of hydroxyl groups with an increase of film thickness; but with a further growth of thickness a decrease in the amount of OH⁻ groups on the film surface was observed. Ennaceri et al. [31] reported gradually increase in the amount of surface OH⁻ groups with the increase in the thickness of the TiO₂ thin films deposited by ultrasonic spray pyrolysis. The results of present study disagree with the observation study of Ennaceri et al. [31] because the TiO₂ thin film deposited at 450 °C is thicker than the sample deposited at 350 °C, showing the high influence of deposition temperature on the surface chemical features. Furthermore, following the [Vo]/[Ti-O] ratios, the amount of oxygen vacancies on the surface of TiO₂ films increased with the increase in the deposition temperature from 250 to 350 °C (Table 2). It was found that the highest amount of oxygen vacancies on the TiO₂ film surface belongs to the film fabricated at 350 °C (Table 2). The lower number of oxygen vacancies defects on TiO₂ thin film deposited at 450 °C could be attributed to the higher deposition temperature: the intensive heat treatment could repair the oxygen vacancy defects. Liu et al. [32] studied TiO₂ thin films deposited by ultrasonic spray pyrolysis and annealed at different temperatures; a decrease in oxygen surface defects on TiO₂ thin film with an increase in annealing temperature was reported.”

Page 7, 5th paragraph:

“Thus, XPS results showed that the film deposited at 350 °C has the highest amount of OH⁻ groups and oxygen vacancy defects and less carbon content on the surface of the film (Table 2). The OH⁻ on TiO₂ surface are considered very effective for photocatalytic degradation of organic pollutants as they are generally the precursors of hydroxyl radicals. In addition, the surface oxygen vacancy defects are crucial because water can dissociate on oxygen vacancies with the formation of two bridging OH⁻ groups creating more OH⁻ groups on the film surface, which is beneficial for adsorbing VOCs by forming hydrogen bonds with functional groups [28, 33]. The carbon impurities were found giving an inhibiting effect to the active sites on the surface of TiO₂ thin films, and thus decreasing the photocatalytic activity [14].”

Page 7, 8th paragraph:

“Additionally, the presence of the surface oxygen defects enhance the wetting properties, leading to the trapping of OH⁻ groups [28]. Simonsen et al. [30] studied the effect of OH⁻ groups on superhydrophilic properties of TiO₂ thin films with different characteristics: a linear correlation between the OH⁻ amount on the surface of the film and the water CA was observed. On the other hand, Chen et al. reported that the TiO₂ thin film which has the lowest level of OH⁻ and Vo, also demonstrated the superhydrophilic property under UV light [28].”

4.5. Photocatalytic Activity

Page 9, 7th paragraph:

“In the present study, the thickness of TiO₂ thin films increased with an increase to the deposition temperature (Table 1). It has been reported [44] that higher photocatalytic activity is attributed to a rougher surface of the film with higher thickness. However, in several studies, it has been reported that with the further increase in the thickness of films, the length of the migration path of the carriers to the surface of the catalyst increases while their generation rate remains constant; the charge carriers experience higher recombination rates, resulting in an overall decrease in the photocatalytic activity [44]. This could be another reason of lower photocatalytic activity of the TiO₂ thin film deposited at 450 °C if compared to the TiO₂ film deposited at 350 °C.”

Page 9, 8th paragraph:

“Usually, it is quite difficult to compare the photocatalytic performance of thin films with that of coatings obtained from nanoparticles. The specific quantity of TiO₂ expressed as mg of titania per cm² of surface area of the coating could serve for a proper comparison of different studies regarding the photocatalytic oxidation of MTBE. In the present study, the specific quantity of TiO₂ in the film deposited at 350 °C was around 0.2 mg cm⁻², achieving 80% conversion of MTBE on a coated window glass with the surface area of 600 cm². Preis et al. [6] reported the specific quantity of TiO₂ (P25) in the coated reactor (640 cm²) as 1.4 mg cm⁻². The 30% conversion of MTBE (inlet concentration of 100 ppm) with acetone detected as the only gas-phase by-product was observed in this reactor (the reactor temperature was 59.8 °C). Galanos et al. [5] indicated the specific quantity of TiO₂ in the coated reactor as 3.5 mg cm⁻². This TiO₂ coating showed 90% of MTBE (500 ppm) conversion in up to 36.6 sec. of residence time. Acetone and TBF were the only intermediate by-products detected during the photocatalytic reaction. Park et al. [4] presented the calculated reaction rate values during the photocatalytic degradation of MTBE as a function of different specific

quantities of TiO₂. It has been found that the highest reaction rate (0.85 μmol min⁻¹) was obtained with the TiO₂ (P25) coating of 0.6 mg cm⁻² and that the reaction rate stayed constant even after the increase in the specific catalyst quantity.”

Comment 6: Explain the increase and the decrease of grain size.

Answer:

The authors would like to thank the referee for his/her valuable remark. The grain size is influenced by the deposition temperature of the TiO₂ films. As seen from SEM images Fig 1, the grain size is increasing with increasing deposition temperature. While the surface of the film deposited at 250 °C and annealed at 500 °C consists of grains with a size of ca 20 nm, the film deposited at 450 °C and annealed at 500 °C consists of grains with a size of ca 50 nm. However, an increase and a decrease were observed in the mean crystallite size. Even though the annealing temperature (500 °C) is same, the crystallite size are different regarding to the differences in growing temperature. There is an increase in the mean crystallite size from 13 nm to 35 nm when the deposition temperature increases from 250 to 350 °C, while the TiO₂ film deposited at 450 °C showed 32 nm. The slight decrease of ca. 3 nm could be attributed to the fitting error, which is ca 10%.

Comment 7: How much the thickness values of the films?

Answer: Thank you for the question. We agree with referee that thickness optimization is needed. In our studies, we kept all deposition parameters constant, except deposition temperature. Therefore, we can relate film thickness and specific surface properties to the specific deposition temperature.

It now reads in the revised version on:

Page 9, 8th paragraph:

“In the present study, the thickness of TiO₂ thin films increased with an increase to the deposition temperature (Table 1). It has been reported [44] that higher photocatalytic activity is attributed to a rougher surface of the film with higher thickness. However, in several studies, it has been reported that with the further increase in the thickness of films, the length of the migration path of the carriers to the surface of the catalyst increases while their generation rate remains constant; the charge carriers experience higher recombination rates, resulting in an overall decrease in the photocatalytic activity [44]. This could be another reason of lower photocatalytic activity of the TiO₂ thin film deposited at 450 °C if compared to the TiO₂ film deposited at 350 °C.”

Comment 8: The English grammar should be reviewed.

Answer:

The English was polished according to the remark of the Reviewer.

We appreciate very much for the valuable comments by the Reviewers and hope that the revised manuscript would now be acceptable for printing in ROS.

All changes made are highlighted in the revised version of the manuscript: RSOS-181578_revised_highlighted.docx

On behalf of co-authors,

I. Dündar

I. Oja Acik

Corresponding author